# TRAINING NEURAL NETWORKS TO OPERATE AT HIGH ACCURACY AND LOW MANUAL EFFORT

## ABSTRACT

In human-AI collaboration systems for critical applications based on neural networks, humans should set an operating point based on a model's confidence to determine when the decision should be delegated to experts. The underlying assumption is that the network's confident predictions are also correct. However, modern neural networks are notoriously overconfident in their predictions, thus they achieve lower accuracy even when operated at high confidence. Network calibration methods mitigate this problem by encouraging models to make predictions whose confidence is consistent with the accuracy, i.e., encourage confidence to reflect the number of mistakes the network is expected to make. However, they do not consider that data need to be manually analysed by experts in critical applications if the confidence of the network is below a certain level. This can be crucial for applications where available expert time is limited and expensive, e.g., medical ones. The trade-off between the accuracy of the network and the amount of samples delegated to expert at every confidence threshold can be represented by a curve. In this paper we propose a new loss function for classification that takes into account both aspects by optimizing the area under this curve. We perform extensive experiments on multiple computer vision and medical image datasets for classification and compare the proposed approach with the existing network calibration methods. Our results demonstrate that our method improves classification accuracy while delegating less number of decisions to human experts, achieves better out-of-distribution samples detection and on par calibration performance compared to existing methods.

## 1 INTRODUCTION

Artificial intelligence (AI) systems based on deep neural networks have achieved state-of-the-art results by reaching or even outperforming human-level performance in many predictive tasks Esteva et al. (2017); Rajpurkar et al. (2018); Chen et al. (2017); Szegedy et al. (2016). Despite the great potential of neural networks for automating various tasks, there are pitfalls when they are used in a fully automated setting, which makes them difficult to deploy in safety-critical applications such as healthcare Kelly et al. (2019); Quinonero-Candela et al. (2008); Sangalli et al. (2021). Human-AI collaboration aims at tackling these issues by keeping humans in the loop and building systems that take advantage of humans and AI by minimizing their shortcomings Patel et al. (2019).

A simple way of building collaboration between a network and a human expert would be delegating the decisions to the expert when the network's confidence score is lower than a predetermined threshold which we refer to as "operating point". For example, in healthcare, a neural network trained for predicting whether a lesion is benign or malignant should leave the decision to the human doctor if not very confident Jiang et al. (2012). In such cases the domain knowledge of the doctor could be exploited to assess more ambiguous cases, where for example education or previous experience can play a crucial role in the evaluation. Another example of human-AI collaboration is hate speech detection for social media (Conneau & Lample, 2019), where neural networks highly reduce the load of manual analysis of contents required by humans. In industrial systems, curves are employed (Gorski et al., 2001) that assess a predictive model in terms of accuracy and number of samples that requires manual assessment from a human expert for varying operating points that loosely relate to varying confidence levels of the algorithm's prediction. We will refer to this performance curve as *Confidence Operating Characteristics (COC)*, as it reminds of the classic Re-

ceiver Operating Characteristic (ROC) curve where an analogous balance is sought after between Sensitivity and Specificity of a predictive model. COC curve can be used by domain experts, such as doctors, to identify the most suitable operating point that balances *performance and amount of data to re-examine* for the specific task. The underlying assumption in these applications is that the confidence level of networks indicates when the predictions are likely to be correct or incorrect. However, modern deep neural networks that achieve state-of-the-art results are known to be over-confident even in their wrong predictions. This leads to networks that are not well-calibrated, i.e., the confidence scores do not properly indicate the likelihood of the correctness of the predictions Guo et al. (2017). Thus, neural networks suffer from lower accuracy than expected, when operated at high confidence thresholds.

Network calibration methods mitigate this problem by calibrating the output confidences of the model Guo et al. (2017); Kumar et al. (2018); Karandikar et al. (2021); Gupta et al. (2021). However, they do not consider that data may need to be manually analyzed by experts in critical applications if the confidence of the network is below a certain level. This can be crucial for various applications where expert time is limited and expensive. For example, in medical imaging, the interpretation of more complex data requires clinical expertise and the number of available experts is extremely limited, especially in low-income countries Kelly et al. (2019). This motivates us to take the expert load into account along with accuracy when assessing the performance of human-AI collaboration systems and training neural networks.

In this paper, we make the following contributions:

- We propose a new loss function for multi-class classification that takes into account both of the aspects by maximizing the area under COC (AUCOC) curve.
- We perform experiments on two computer vision and one medical image datasets for multi-class class classification. We compare the proposed AUCOC loss with the conventional loss functions for training neural networks as well as network calibration methods. The results demonstrate that our method improves other methods in terms of both accuracy and AUCOC.
- We evaluate network calibration and out-of-distribution (OOD) samples detection performance of all methods. The results show that the proposed approach is able to consistently achieve better OOD samples detection and on par network calibration performance.

## 1.1 RELATED WORK

In industrial applications, curves that plot network accuracy on accepted samples against manual workload of a human expert are used for performance analysis of the system (Gorski et al., 2001). To the best of our knowledge this is the first work that explicitly takes into account during the optimization process the trade-off between neural network performance and amount of data to be analysed by a human expert, in a human-AI collaborative system. Therefore, there is no direct literature that we can compare with. We found the literature on network calibration methods the closest to our setting because they also aim at improving the interaction between human and AI, by enabling the networks to delegate the decision to human when they are not very confident. Therefore, we compare our method with the existing network calibration methods in the literature.

In a well-calibrated network, the probability associated with the predicted class label should reflect the likelihood of the correctness of the predictions. Guo et al. (2017) defines the calibration error as the difference in expectation between accuracy and confidence in each confidence bin. One category of calibration methods augments or replaces the conventional training losses with another loss to explicitly encourage reducing the calibration error. Kumar et al. (2018) propose a method called MMCE loss by replacing the bins with continuous kernel to obtain a continuous distribution and a differentiable measure of calibration. Karandikar et al. (2021) propose two loss functions for calibration, called Soft-AvUC and Soft-ECE, by replacing the hard confidence thresholding in AvUC (Krishnan & Tickoo, 2020) and binning in ECE Guo et al. (2017) with smooth functions, respectively. All these three functions are used as a secondary losses along with conventional losses such as cross-entropy. Mukhoti et al. (2020) find that Focal Loss (FL) (Lin et al., 2017) provides inherently more calibrated models, even if it was not originally designed to improve calibration, as it adds implicit weight regularisation. The second category of methods are post-hoc calibration approaches, which rescale model predictions after training. Platt scaling (Platt, 2000) and histogram

binning (Zadrozny & Elkan, 2001) fall into this class. Temperature scaling (TS) (Guo et al., 2017), a modern variant of Platt scaling, is the most popular approach of this group. The idea of TS is to scale the logits of a neural network by dividing by a positive scalar such that they do not saturate after the subsequent softmax or sigmoid activation. TS can be used as a complementary method for all networks and it does not affect the accuracy of the models while significantly improving calibration.

## 2 METHODS

In this section, we illustrate more in detail the curve that assesses a predictive model in terms of accuracy and the number of samples that are delegated to human expert for manual analysis, that has been used in industry, e.g., (Gorski et al., 2001). We will refer to it as to Confidence Operating Characteristics (COC), as it reminds of the classic Receiver Operating Characteristic (ROC) curve where an analogous balance is sought after Sensitivity and Specificity of a predictive model. Then, we describe a novel cost function for training of neural networks: *the area under COC (AUCOC) loss (AUCOCLoss)*.

### 2.1 NOTATION

Let $D = \langle (x_n, y_n) \rangle_{n=1}^N$ denote a dataset composed of $N$ samples from a joint distribution $\mathcal{D}(\mathcal{X}, \mathcal{Y})$, where $x_n \in \mathcal{X}$ and $y_n \in \mathcal{Y} = \{1, 2, ..., K\}$ denote the input data and the corresponding class label, respectively. Let $f_\theta(y|x)$ be the probability distribution predicted by a neural network $f$ parametrized by $\theta$ for an input $x$. For each data point $x_n$, $\hat{y} = \mathrm{argmax}_{y \in \mathcal{Y}} f_\theta(y|x_n)$ denotes the predicted class label, associated to a *correctness score* $c_n = \mathbb{1}(\hat{y}_n = y_n)$ and to a *confidence score* $r_n = \max f_\theta(y|x_n)$, where $r_n \in [0, 1]$ and $\mathbb{1}(.)$ is an indicator function. $\mathbf{r} = [r_1, ... r_N]$ represents the vector containing all the predicted confidences for a set of data points. $p(r)$ denotes the probability distribution over $r$ values. In a human-AI collaboration, samples with confidence $r$ lower than a threshold $r_0$ would be delegated to a human expert for manual assessment.

### 2.2 CONFIDENCE OPERATING CHARACTERISTICS (COC) CURVE

Our first goal is to find an appropriate evaluation method to assess the trade-off between a neural network's performance and the number of samples that requires manual analysis from a domain expert in a human-AI collaboration system. We decide to optimize the area under COC as it provides practitioners with flexibility in the choice of the operating point, similarly to classic ROC curve.

**x-y axes of COC:** In order to construct the COC curve, first, we define a sliding threshold $r_0$ over the space of predicted confidences $r$ and assume that the samples with confidence scores lower than $r_0$ are given to a human expert for decision. Then, for each threshold $r_0$, we calculate ($x-$axis) the percentage of samples that are delegated to human expert and ($y-$axis) the accuracy of the network on the remaining samples corresponding to the threshold $r_0$.

We formulate the axes of the COC curve as follows

$$x - \text{axis}: \ \tau_0 = p(r < r_0) = \int_0^{r_0} p(r) dr \qquad y - \text{axis}: \ \mathbb{E}[c|r \geq r_0] \qquad (1)$$

For each threshold level $r_0$, $\tau_0$ represents the fraction of samples whose confidence is lower than that threshold, i.e., the percentage of the samples that are delegated to expert for manual analysis. $\mathbb{E}[c|r \geq r_0]$ corresponds to the expected value of the correctness score $c$ for all the samples for which the network's confidence is equal or larger than the threshold $r_0$, i.e., the samples for which network prediction will be used. This expected value can be computed as

$$\mathbb{E}[c|r \geq r_0] = \int_0^1 \mathbb{E}[c|r] p(r) dr / (1 - \tau_0). \qquad (2)$$

We provide the derivation of Eq. 2 in Appendix A.1.

The area under COC curve (AUCOC), like the area under ROC curve, is a global indicator of the performance of a system. Higher AUCOC indicates lower number of samples that are delegated to human experts or/and higher accuracy for the samples that are analysed by the network. Lower AUCOC on the other hand, indicates higher number of delegations to human experts or/and lower

accuracy. AUCOC can be computed by integrating the $y-$axis expressed in Eq. 1 over the whole range of $\tau_0 \in [0, 1]$:

$$AUCOC = \int_0^1 \mathbb{E}[c|r \geq r_0]d\tau_0 = \int_0^1 \left\{ \int_{r_0}^1 \mathbb{E}[c|r]p(r)dr \right\} \frac{d\tau_0}{1 - \tau_0} \tag{3}$$

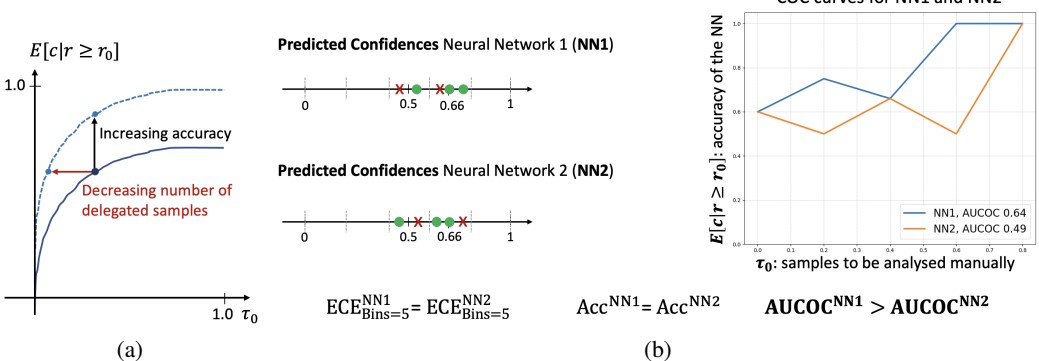

Figure 1: **(a)** shows how one could improve AUCOC, increasing the accuracy of the network and/or decreasing the amount of data to be analysed by the domain expert. The dashed curve has higher AUCOC than the solid one. **(b)** illustrates a toy example where two models have the same accuracy and ECE with 5 bins. However, they have different AUCOC values due to different ordering of correctly and incorrectly classified samples according to the assigned confidence by the network.

## 2.3 AUCOCLOSS: MAXIMIZING AUCOC FOR TRAINING NEURAL NETWORKS

In the previous section, we described COC to assess the performance of a system in terms of accuracy and number of samples delegated to human experts at various operating points and mentioned that higher AUCOC is desired. With this motivation, in this section, we introduce a new loss function called AUCOCLoss that maximizes AUCOC for training neural networks. AUCOC's explicit maximization would enforce the reduction of the number of samples delegated to human expert while maintaining the accuracy level assessed by the algorithm (i.e., keeping $\mathbb{E}[c|r \geq r_0]$ constant) and/or the improvement in the prediction accuracy of the samples analysed by the algorithm while maintaining a particular amount of data to be analysed by the human (i.e., keeping $\tau_0$ constant) as illustrated in Figure 1a.

We define our loss function to maximize AUCOC as

$$AUCOCLoss = -\log(AUCOC). \tag{4}$$

We use the negative logarithm as AUCOC lies in the interval $[0, 1]$, which corresponds to AU-COCLoss $\in [0, \inf]$ which is suitable for minimizing cost functions. For computing AUCOC for training, we use kernel density estimation (KDE) with training samples to estimate $p(r)$ in Eq. 3

$$p(r) \approx \frac{1}{N} \sum_{n=1}^N K(r - r_n) \tag{5}$$

where $K$ is a Gaussian kernel and we estimate its bandwidth using Scott's rule of thumb Scott (1979). Then, the other terms in Eq. 3, $\mathbb{E}[c|r]p(r)$ and $\tau_0$, are estimated as follows

$$\mathbb{E}[c|r]p(r) \approx \frac{1}{N} \sum_{n=1}^N c_n K(r - r_n) \qquad \tau_0 \approx \frac{1}{N} \int_0^{r_0} \sum_{n=1}^N K(r - r_n)dr. \tag{6}$$

## 2.4 TOY EXAMPLE: ADDED VALUE BY COC CURVE AND AUCOC

In this section, we demonstrate the added value of assessing the performance of a predictive model using COC curve and AUCOC compared to expected calibration error (ECE) Guo et al. (2017) and classification accuracy on a toy example.

The toy example on the left side of Figure 1b shows the confidence scores of 5 samples from two different models, NN1 and NN2, for a classification problem. The green circles denote the predicted confidences for correctly classified samples, while the red crosses the confidences of the misclassified ones. ECE divides the confidence space into bins, computes the difference between the average accuracy and confidence scores for each bin, and returns the average of the differences as the final measure of calibration error. Let us assume that we divide the confidence space into 5-bins as indicated with the gray dotted lines in the confidence spaces of NN1 and NN2 for computing ECE. This results in equal ECE for both networks. Note that these models also have equal classification accuracy since each classifies 2 out of 5 samples incorrectly. Looking at these two performance metrics, it is not possible to choose one model over the other since NN1 and NN2 perform identically.

On the contrary, the AUCOC is larger for NN1 than for NN2, as shown in the rightmost plot in Figure 1b. The difference in AUCOC is due to the different ordering of correctly and incorrectly classified samples, which COC and AUCOC are able to detect. By looking at the AUCOC results, one would prefer NN1 compared to NN2. Indeed, NN1 is a better model than NN2 because it achieves either equal or better accuracy than NN2, for the same amount of data to be manually examined. Analogously, it delegates either equal or lower number of samples to expert for the same accuracy level.

### 2.5 IMPLEMENTATION DETAILS

**COC:** We implement the selection of the operating points in COC by thresholding on the predicted output confidences $\mathbf{r}$. First, we arrange the confidences $\mathbf{r}$ of the whole dataset (or batch) in ascending order. Each predicted confidence is then selected as threshold level for the vector $\mathbf{r}$, and $\tau_0$ and $\mathbb{E}[c|r \geq r_0]$ are computed. This allows us to save time on threshold selections compared to exploring the confidence space with arbitrarily fine-grained levels.

**AUCOCLoss:** There are two important points about the implementation of AUCOCLoss. First, instead of using $\mathbb{E}[c|r]p(r) \approx \frac{1}{N}\sum_{n=1}^{N} c_n K(\|r - r_n\|)$ as given in Eq. 6, we approximated it as $\mathbb{E}[c|r]p(r) \approx \frac{1}{N}\sum_{n=1}^{N} r^* K(\|r - r_n\|)$ where $r_n^* = f_\theta(y_n|x_n)$ is the confidence of the correct class for a sample $n$. The main reason of this modification is that the gradient of the misclassified samples becomes zero because $c_n$ is zero when a sample $x_n$ is not classified correctly. Therefore, the network cannot learn how to classify samples correctly, once they are misclassified. To deal with this issue, we replaced the correctness score $c_n$, which can be either 0 or 1, with $r_n^*$ which can take continuous values between 0 and 1. With this new approximation, we can also compute and back-propagate the gradients for misclassified samples.

## 3 EXPERIMENTS

In this section, we present our experimental evaluations on multi-class image classification tasks. We performed experiments on three datasets: CIFAR100 (Krizhevsky, 2009), Tiny-ImageNet which is a subset of ImageNet (Deng et al., 2009) and on DermaMNIST (Yang et al., 2021). We compared AUCOCLoss with different loss functions, most of which are designed to improve calibration performance, while preserving accuracy: cross-entropy (CE), focal-loss (FL) (Lin et al., 2017), adaptive focal-loss (Mukhoti et al., 2020), maximum mean calibration error loss (MMCE) (Kumar et al., 2018), soft binning calibration objective (Soft-ECE) and soft accuracy versus uncertainty calibration (Soft-AvUC) (Karandikar et al., 2021). We optimized MMCE, Soft-ECE, and Soft-AvUC losses jointly with a primary loss which we used either CE or FL, consistently with the literature (Karandikar et al., 2021). We also performed experiments by optimizing the proposed AUCOCLoss as both primary and secondary loss with CE and FL. KDE in AUCOCLoss is applied batch-wise during the training. In addition, for all the experiments we report results after applying temperature scaling (TS) (Guo et al., 2017) as post-training technique.

We use three metrics to evaluate the performance of the methods: classification accuracy, equal-mass expected calibration error (ECE) (Nixon et al., 2019) with 15 bins, and AUCOC. Classicification accuracy is simply the ratio between number of correct samples over total number of samples and we AUCOC is computed using Eq. 3. Equal-mass ECE divides the confidence space into M bins such that each bin $B_m$ contains equal number of samples and computes weighted average of the difference between the average accuracy and confidence of each bin. In addition, we report some

examples of operating points of COC curve - given a certain accuracy we show the corresponding percentage of samples that need to be analyzed manually ($\tau_0$ @acc) on the COC curves.

A reliable AI system should have lower confidence whenever it encounters out-of-distribution (OOD) data, deferring those samples to the human expert for further investigation. Evaluating the OOD samples detection performance is a common experiment in the network calibration literature. Therefore, we investigated the OOD detection performance of all the methods when CIFAR100-C (Hendrycks & Dietterich, 2019) (with Gaussian noise) and SVHN (Netzer et al., 2011) are OOD datasets, while the network is trained on CIFAR100 (in-distribution). We evaluated the OOD detection performance of all methods using Area Under the Receiver Operating Characteristics (AUROC) curve.

All the experiments were run 3 times and we report the average results in Tables 1, 2 3, and 4.

### 3.1 SETUP DETAILS

For CIFAR100 (Krizhevsky, 2009) we used 45000/5000/10000 images respectively as training/validation/test sets and Wide-Resnet-28-10 (Zagoruyko & Komodakis, 2016), consistently with Karandikar et al. (2021). We trained the model for 200 epochs, using Stochastic Gradient Descent (SGD), with batch of 512, momentum of 0.9 and an initial learning rate of 0.1, decreased after 60, 120, 160 epochs by a factor of 0.1.

Tiny-ImageNet is a subset of ImageNet (Deng et al., 2009) with $64 \times 64$ images and 200 classes. We employed 90000/10000/10000 images as training/validation/test set, respectively. As in Mukhoti et al. (2020) we used ResNet-50 (He et al., 2016) as backbone architecture. SGD was used as optimiser with a batch size of 512, momentum of 0.9 and base learning rate of 0.1, divided by 0.1 at 40th and 60th epochs.

We chose a medical imaging dataset, DermaMNIST (Yang et al., 2021), as the third dataset since human-AI collaboration systems are particularly useful for medical domain. DermaMNIST is composed of dermatoscopic images with 7007/1003/2005 samples for training, validation and test set, respectively, categorised in 7 different diseases. We followed the training procedures of the original paper, employing a ResNet-50 He et al. (2016), Adam optimizer and a batch size of 128. We found that a more suitable initial learning rate is 0.001 and trained the model for 100 epochs, reducing the learning rate by 0.1 after 50 and 75 epochs.

We set the weighting factor for AUCOCLoss when it is used as secondary loss selected via cross-validation. For the baselines we used the hyperparameter combinations specified by the original papers if done so. If not, we carried out cross-validation and selected the settings that provided the best performance on the validation set. In order not to be biased towards a specific metric, for all the methods, we saved the models with respect to the best ECE, accuracy, and AUCOC. We found empirically that models check-pointed using ECE provided very poor results and therefore we omit them while presenting results. Networks check-pointed using either accuracy or AUCOC provided comparable outcome. For Tiny-Imagenet, being the largest dataset, we report results for both check-pointing strategies in Sec. 4. For CIFAR100, DermaMNIST and OOD detection experiments, we only report the AUCOC checkpointing results in the main paper in Sec. 4 and the remaining results in Appendix E.

## 4 RESULTS

In Tables 1 2 and 3 we report the results on CIFAR100, Tiny-Imagenet and DermaMNIST, respectively. For Tiny-Imagenet, the largest dataset, the results for both AUCOC-based and accuracy-based checkpoints are reported, while for the other datasets we reported results only for AUCOC checkpoints in the main paper and the rest in Appendix E. Bold results indicate the methods that performed best for each metric. ↑ means the higher the better for a metric, while ↓ the lower the better.

We empirically found that the optimization of AUCOC alone can be complex and AUCOCLoss mainly provides the best results when it is used as a secondary loss, regularized by another cost function. We find that AUCOCLoss outperformed consistently all the baselines on every dataset in terms of accuracy and AUCOC. In terms of ECE, the proposed loss function provided on par performance compared to the other loss functions particularly designed for network calibration. We

Table 1: Test results on CIFAR100 for accuracy, AUCOC and ECE. For each loss function, we report the results of the model check-pointed on the best AUCOC on validation set, pre and post TS. The last two columns report $\tau_0$ corresponding to 90% and 95% accuracy (pre TS). In bold the best result for each metric. The average results over 3 runs are reported.

| | Pre TS | | | Post TS | | $\tau_0 \downarrow$ @ acc. | |
| Loss funct. | AUCOC $\uparrow$ | Accuracy $\uparrow$ | ECE $\downarrow$ | AUCOC $\uparrow$ | ECE $\downarrow$ | 90% | 95% |
|---|---|---|---|---|---|---|---|
| CE | 91,43 | 75,71 | 11,56 | 92,98 | 2,41 | 29,03 | 44,61 |
| FL ($\gamma$=3) | 93,91 | 77,83 | 4,05 | 93,82 | 1,92 | 24,71 | 40,48 |
| AdaFL53 | 93,89 | 77,64 | 4,15 | 93,79 | 1,47 | 25,08 | 40,74 |
| CE+MMCE | 92,42 | 75,35 | 9,99 | 92,7 | 2,32 | 30,01 | 44,71 |
| FL+MMCE | 93,9 | 77,78 | **1,87** | 93,92 | 1,92 | 25,62 | 40,9 |
| CE+Soft-AvUC | 93,99 | 77,65 | 6,81 | 93,87 | 3,23 | 24,62 | 45.0 |
| FL+ Soft-AvUC | 93,97 | 77,64 | 1,89 | 93,93 | 1,83 | 25,97 | 40,82 |
| CE+Soft-ECE | 93,88 | 77,57 | 7,09 | 93,75 | 3,85 | 24,76 | 40,14 |
| FL+Soft-ECE | 93,41 | 76,69 | 6,43 | 93,56 | 1,74 | 28,13 | 43,29 |
| **CE+AUCOCLoss** | **94,49** | **78,94** | 6,7 | **94,35** | 1,65 | **21,6** | 36,73 |
| **FL+AUCOCLoss** | 94,18 | 78,31 | 2,85 | 94,24 | **1,34** | 23,5 | **36,68** |
| **AUCOCLoss Prim.** | 94,24 | 78,32 | 5,92 | 94,32 | 2,16 | 23,65 | 38,3 |

also observed that the largest improvement in ECE is obtained after temperature scaling (TS) in all methods.

In order to have an idea of what a certain AUCOC corresponds to in practice, Figure 2 show some examples of COC curves (more results are available in Appendix E). Overall, the plots AUCOCLoss lie above all the baselines, which is a desirable behavior as it corresponds to better operating points. Moreover, in Tables 1, 2, 3 we report some operating points of the COC curves for each model. In particular, we measured the percentage of samples delegated to expert ($\tau_0$) at 90% and 95% accuracy for CIFAR100 and DermaMNIST, and at 65% and 75% accuracy for Tiny-Imagenet (as the initial accuracy is also much lower in this dataset). In all the experiments, to varying degrees, AUCOCLoss is able to provide lower delegated samples than the baselines.

In OOD experiments reported in Table 4, we used the model trained on CIFAR100 and evaluated the OOD detection performance on both CIFAR100-C and SVHN dataset. We used the confidence scores of the models with and without applying TS when determining where a sample is OOD or in-distribution. The bold results highlight the best results in terms of AUROC. On both OOD datasets, AUCOCLoss provided the highest AUROC, both before and after applying temperature scaling..

Finally, we investigated how the performances of the models vary when the batch size varies, which may be crucial for KDE-based methods like AUCOCLoss as it may affect the accuracy of density estimation. We investigated this by reducing the batch size from 128 (used in the main experiments) to 64 and 32 in DermaMNIST dataset. We report the results in Table 9 in Appendix E for space reasons. We observe that reducing the batch does not significantly affect the performance of our method.

## 5 CONCLUSION

In this paper we propose a new cost function for multi-class classification that takes into account the trade-off between a neural network's performance and the amount of data that requires manual analysis from a domain expert when the network is not confident enough, by maximizing the area under COC (AUCOC) curve. Extensive experiments on various computer vision and medical image datasets are presented, comparing the new loss with various baselines. The results demonstrate that our approach improves the other methods in terms of both accuracy and AUCOC and provides comparable ECE. Additionally, evaluate the performance of different losses for OOD samples detection and show that our method outperforms the baselines.

While we presented COC and the loss based on AUCOC for multi-class classification, extensions to other tasks is possible and we will explore this direction in future work. Another possible direction would be investigating other performance metrics to embed in the $y$-axis of COC other than accuracy.

Table 2: Test results on Tiny-ImageNet for accuracy, AUCOC and ECE. For each loss function, the first row reports the results of the model check-pointed on the best AUCOC (in gray) and the second row on the best accuracy on validation set, pre and post TS. The last two columns report $\tau_0$ corresponding to 65% and 75% accuracy (pre TS). In bold the best result for each metric. The average results over 3 runs are reported.

| Loss funct. | Pre TS | | | Post TS | | $\tau_0 \downarrow$ @ acc. | |
| --- | --- | --- | --- | --- | --- | --- | --- |
| | AUCOC $\uparrow$ | Accuracy $\uparrow$ | ECE $\downarrow$ | AUCOC $\uparrow$ | ECE $\downarrow$ | 65% | 75% |
| CE | 72,56 | 47,39 | 21,39 | 72,99 | 1,54 | 39,88 | 56,29 |
| | 72,68 | 47,62 | 22,04 | 73,07 | 1,97 | 39,35 | 56,03 |
| FL ($\gamma$=3) | 73,12 | 47,71 | 10,33 | 73,09 | 1,46 | 38,4 | 55,08 |
| | 73,05 | 47,61 | 10,72 | 73,02 | 1,61 | 38,71 | 55,15 |
| AdaFL53 | 73,19 | 47,81 | 10,98 | 73,19 | 1,35 | 38,56 | 55,2 |
| | 73,21 | 47,78 | 10,47 | 73,18 | 1,52 | 38,33 | 55,38 |
| CE+MMCE | 72,47 | 47,11 | 21,12 | 72,73 | 1,79 | 39,94 | 56,7 |
| | 72,5 | 47,04 | 20,76 | 72,69 | 1,65 | 39,99 | 56,79 |
| FL+MMCE | 73,51 | 48,03 | 2,82 | 73,47 | 1,94 | 37,83 | 55,15 |
| | 73,44 | 47,89 | 3,62 | 73,41 | 1,62 | 37,78 | 55,35 |
| CE+Soft-AvUC | 72,92 | 47,89 | 21,77 | 73,47 | 2,13 | 38,28 | 54,91 |
| | 72,89 | 47,89 | 21,7 | 73,44 | 1,87 | 38,41 | 55,08 |
| FL+ Soft-AvUC | 74,3 | 48,69 | 7,18 | 74,34 | **1,51** | 35,83 | 53,21 |
| | 73,23 | 48,15 | 22,01 | 73,81 | 1,86 | 37,93 | 54,19 |
| CE+Soft-ECE | 72,94 | 47,65 | 17,56 | 72,95 | 1,66 | 38,81 | 55,84 |
| | 72,94 | 47,65 | 17,56 | 72,95 | 1,66 | 38,81 | 55,84 |
| FL+Soft-ECE | 72,61 | 47,4 | 1,964 | 72,6 | 2,6 | 39,94 | 56,71 |
| | 72,61 | 47,39 | **2,99** | 72,59 | 2,1 | 39,71 | 56,69 |
| **CE+AUCOCLoss** | 74,56 | 49,1 | 12,33 | 74,66 | 1,65 | 34,78 | **52,01** |
| | **74,74** | **49,21** | 12,48 | **74,7** | 1,67 | **34,62** | 52,28 |
| **FL+AUCOCLoss** | 74,3 | 49,19 | 3,23 | 74,25 | 1,86 | 34,85 | 53,15 |
| | 74,3 | 49,12 | 3,17 | 74,27 | 2,06 | 34,86 | 53,01 |
| **AUCOCLoss Prim.** | 73.03 | 47.7 | 8.18 | 73,22 | 2,5 | 37,93 | 55,64 |
| | 72,89 | 47,5 | 8,13 | 73,08 | 2,58 | 38,49 | 55,76 |

Table 3: Test results on DermaMNIST for accuracy, AUCOC and ECE. For each loss function, we report the results of the model check-pointed on the best AUCOC on validation set, pre and post TS. The last two columns report $\tau_0$ corresponding to 90% and 95% accuracy (pre TS). In bold the best result for each metric. The average results over 3 runs are reported.

| Loss funct. | Pre TS | | | Post TS | | $\tau_0 \downarrow$ @ acc. | |
| --- | --- | --- | --- | --- | --- | --- | --- |
| | AUCOC $\uparrow$ | Accuracy $\uparrow$ | ECE $\downarrow$ | AUCOC $\uparrow$ | ECE $\downarrow$ | 90% | 95% |
| CE | 89,84 | 71,59 | 8,6 | 90,17 | **3,07** | 43,51 | 56,21 |
| FL ($\gamma$=3) | 90,5 | 72,64 | 11,97 | 90,94 | 4,24 | 40,63 | 53,9 |
| AdaFL53 | 90,11 | 73,1 | 12,54 | 90,73 | 3,88 | 40,78 | 55,86 |
| CE+MMCE | 89,71 | 70,99 | 10,87 | 89,77 | 3,59 | 45,82 | 58,07 |
| FL+MMCE | 89,34 | 71,72 | 4,35 | 89,57 | 3,65 | 46,81 | 59,1 |
| CE+Soft-AvUC | 89,67 | 71,51 | 5,58 | 90,15 | 3,79 | 43,04 | 57,09 |
| FL+ Soft-AvUC | 89,42 | 71,04 | **4,09** | 89,62 | 3,48 | 45,47 | 58,69 |
| CE+Soft-ECE | 89,54 | 71,46 | 9,35 | 89,91 | 3,23 | 43,48 | 57,82 |
| FL+Soft-ECE | 90,22 | 72,62 | 10,31 | 90,53 | 4,5 | 40,95 | 57,46 |
| **CE+AUCOCLoss** | 90,77 | 74,3 | 12,98 | 91,01 | 5,7 | 39,01 | **52,7** |
| **FL+AUCOCLoss** | **91,35** | **74,8** | 7,69 | **91,44** | 5,1 | **37,3** | 53,9 |
| **AUCOCLoss Prim.** | 90,87 | 73,42 | 9,11 | 90,93 | 8,25 | 40,4 | 55,44 |

We believe that this new direction of considering expert load in human-AI collaborated system is important, and COC curve and AUCOCLoss will serve as a baseline for future work.

Table 4: Test AUROC(%) on OOD detection for models trained on CIFAR100 and tested on CIFAR100-C (Gaussian noise) and SVHN, pre and post TS, for models check-pointed for AUCOC.

| Loss funct. | **CIFAR100-C**, AUROC ↑ | | **SVHN**, AUROC ↑ | |
|---|---|---|---|---|
| | Pre TS | Post TS | Pre TS | Post TS |
| CE | 74,37 | 75,22 | 77,42 | 79,42 |
| FL ($\gamma$=3) | 75,12 | 75,35 | 76,61 | 77,08 |
| AdaFL53 | 74,53 | 74,68 | 80,3 | 81,28 |
| CE+MMCE | 74,67 | 74,53 | 75,79 | 77,39 |
| FL+MMCE | 74,42 | 74,39 | 77,57 | 77,41 |
| CE+Soft-AvUC | 73,63 | 73,62 | 78,04 | 78,9 |
| FL+ Soft-AvUC | 72,78 | 76,72 | 79,68 | 79,68 |
| CE+Soft-ECE | 73,29 | 73,18 | 77,02 | 77,69 |
| FL+Soft-ECE | 74,29 | 73,97 | 79,68 | 81,17 |
| **CE+AUCOCLoss** | 76,03 | 76,8 | **82,03** | **83,5** |
| **FL+AUCOCLoss** | 76,31 | 76,02 | 80,11 | 79,35 |
| **AUCOCLoss Prim.** | **79,02** | **78,86** | 81,12 | 79,82 |

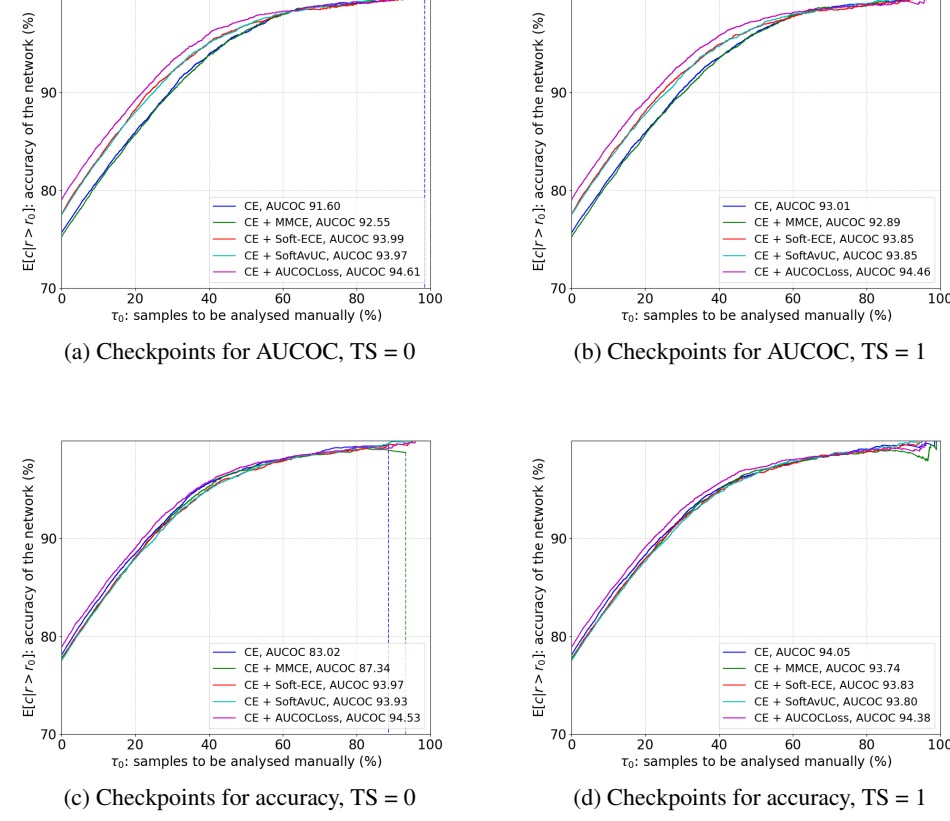

(a) Checkpoints for AUCOC, TS = 0

(b) Checkpoints for AUCOC, TS = 1

(c) Checkpoints for accuracy, TS = 0

(d) Checkpoints for accuracy, TS = 1

Figure 2: COC curves on CIFAR100, run for one seed. In the first row the plots for models check-pointed with AUCOC, the second with accuracy. In the first column models without TS, in the second with TS.

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

## A   APPENDIX

### A.1   DERIVATION OF EQUATION 3

The y-axis in COC curve is expressed mathematically by:

$$
\mathbb{E}[c|r \geq r_0] = \sum p(c|r \geq r_0)c = \sum \frac{p(c, r \geq r_0)}{p(r \geq r_0)}c = \sum \frac{\int_{r_0}^{1} p(c, r)dr}{\int_{r_0}^{1} p(r)dr}c
$$

$$
= \sum \frac{\int_{r_0}^{1} p(c|r)p(r)dr}{\int_{r_0}^{1} p(r)dr}c = \frac{\int_{r_0}^{1} \sum p(c|r)cp(r)dr}{\int_{r_0}^{1} p(r)dr} = \frac{\int_{r_0}^{1} \mathbb{E}[c|r]p(r)dr}{\int_{r_0}^{1} p(r)dr} =
$$

$$
= \frac{\int_{r_0}^{1} \mathbb{E}[c|r]p(r)dr}{1 - \tau_0}
$$

The x-axis in COC curve is expressed mathematically by:

$$
\tau_0 = p(r < r_0) = \int_{0}^{r_0} p(r)dr
$$

Using the $\tau_0$ formulation, we can rewrite the y-axis as

$$
\mathbb{E}[c|r \geq r_0] = \frac{\int_{r_0}^{1} \mathbb{E}[c|r]p(r)dr}{1 - \tau_0}
$$

Maximizing the area under this curve, over $\tau_0 \in [0, 1]$ corresponds to

$$
\max \mathcal{A} = \max \int_{0}^{1} \left\{ \int_{r_0}^{1} \mathbb{E}[c|r]p(r)dr \right\} \frac{d\tau_0}{1 - \tau_0}
$$

Let us assume to use a Gaussian kernel with this expression:

$$
K(\|r - r_n\|) = \frac{1}{\sqrt{2\pi}\alpha} \cdot \exp\left(-\frac{(r - r_n)^2}{2\alpha^2}\right) \tag{7}
$$

Developing the equation, the area calculation becomes:

$$
\mathcal{A} = \int_{0}^{1} \left\{ \int_{r_0}^{1} \mathbb{E}[c|r]p(r)dr \right\} \frac{d\tau_0}{1 - \tau_0} =
$$

$$
= \int_{0}^{1} \left\{ \int_{r_0}^{1} \frac{1}{N} \sum_{n=1}^{N} \mathbf{1}(c_n) K(\|r - r_n\|)dr \right\} \frac{d\tau_0}{1 - \tau_0} =
$$

$$
= \int_{0}^{1} \left\{ \int_{r_0}^{1} \frac{1}{N} \sum_{n=1}^{N} \mathbf{1}(c_n) \frac{1}{\sqrt{2\pi}\alpha} \cdot \exp\left(-\frac{(r - r_n)^2}{2\alpha^2}\right)dr \right\} \frac{d\tau_0}{1 - \tau_0} =
$$

$$
= \int_{0}^{1} \left\{ \frac{1}{N} \sum_{n=1}^{N} \mathbf{1}(c_n) \int_{r_0}^{1} \frac{1}{\sqrt{2\pi}\alpha} \cdot \exp\left(-\frac{(r - r_n)^2}{2\alpha^2}\right)dr \right\} \frac{d\tau_0}{1 - \tau_0} =
$$

$$
= \int_{0}^{1} \left\{ \frac{1}{N} \sum_{n=1}^{N} \mathbf{1}(c_n) \left( ndtr\left(\frac{1 - r_n}{\sqrt{cov}}\right) - ndtr\left(\frac{r_0 - r_n}{\sqrt{cov}}\right) \right) \right\} \frac{d\tau_0}{1 - \tau_0} =
$$

$$
= \sum_{k=1}^{\#thresh} \frac{f(\tau_{0,k-1}, r_{0,k-1}) + f(\tau_{0,k}, r_{0,k})}{2}(\tau_{0,k} - \tau_{0,k-1})
$$

$(8)$

Where:

$$
\tau_{0,k} = \int_{0}^{r_{0,k}} p(r)dr \approx \frac{1}{N} \int_{0}^{r_{0,k}} \sum_{n=1}^{N} K(\|r - r_n\|)dr =
$$

$$
= \frac{1}{N} \sum_{n=1}^{N} \left( ndtr\left(\frac{r_{0,k} - r_n}{\sqrt{cov}}\right) - ndtr\left(\frac{-r_n}{\sqrt{cov}}\right) \right) \tag{9}
$$

Where $ndtr$ expresses the Gaussian cumulative distribution function and the last row in Equation 8 exploits the trapezoidal rule for integrals computation.

## A.2 DERIVATIONS OF THE GRADIENTS OF AUCOC

$$\frac{d}{d\theta}\mathcal{A} = \int_0^1 \frac{d}{d\theta}\left\{\int_{r_0}^1 \mathbb{E}[c|r]p(r)dr\right\}\frac{d\tau_0}{1-\tau_0}$$

Here, we use the assumption discussed in Section 2 that $\tau_0$ does not depend on any parameter, thus allowing us to apply Leibnitz's integration rule, obtaining:

$$\frac{d}{d\theta}\mathcal{A} = \int_0^1 \frac{d}{d\theta}\left\{\int_{r_0}^1 \mathbb{E}[c|r]p(r)dr\right\}\frac{d\tau_0}{1-\tau_0} \tag{10}$$

$$= \int_0^1 \left\{\int_{r_0}^1 \frac{d}{d\theta}\mathbb{E}[c|r]p(r)dr - \mathbb{E}[c|r_0]p(r_0)\frac{dr_0}{d\theta}\right\}\frac{d\tau_0}{1-\tau_0} \tag{11}$$

$\tau_0$ can be expressed as:

$$\tau_0 = p(r \le r_0) = \int_0^{r_0} p(r)dr \tag{12}$$

Consequently:

$$\frac{d\tau_0}{d\theta} = \int_0^{r_0}\frac{dp(r)}{d\theta}dr + p(r_0)\frac{dr_0}{d\theta} = 0$$
$$\frac{dr_0}{d\theta} = -\frac{\int_0^{r_0}\frac{dp(r)}{d\theta}dr}{p(r_0)} \tag{13}$$

Plugging this expression back into Equation 10 we obtain:

$$\frac{d\mathcal{A}}{d\theta} = \int_0^1 \left\{\int_{r_0}^1 \frac{d}{d\theta}\mathbb{E}[c|r]p(r)dr + \mathbb{E}[c|r_0]\int_0^{r_0}\frac{dp(r)}{d\theta}dr\right\}\frac{d\tau_0}{1-\tau_0} \tag{14}$$

Assuming the use of a Gaussian kernel:

$$K(||r - r_n||) = \frac{1}{\sqrt{2\pi}\alpha}\cdot\exp\left(-\frac{(r-r_n)^2}{2\alpha^2}\right) \tag{15}$$

And re-writing:

$$\frac{\mathbb{E}[c|r_0]p(r_0)}{p(r_0)} \approx \frac{\frac{1}{N}\sum_{n=1}^N \mathbf{1}(c_n)K(||r_0 - r_n||)}{\frac{1}{N}\sum_{n=1}^N K(||r_0 - r_n||)} \tag{16}$$

The gradient of the area becomes:

$$\frac{d\mathcal{A}}{dr_n} = \int_0^1 \left\{\int_{r_0}^1 \frac{d}{dr_n}\mathbb{E}[c|r]p(r)dr + \mathbb{E}[c|r_0]\int_0^{r_0}\frac{dp(r)}{dr_n}dr\right\}\frac{d\tau_0}{1-\tau_0} =$$

$$= \int_0^1 \{\int_{r_0}^1 \frac{d}{dr_n}\frac{1}{\sqrt{2\pi}\alpha}\frac{1}{N}\sum_{n=1}^N \mathbf{1}(c_n)\cdot\exp\left(-\frac{(r-r_n)^2}{2\alpha^2}\right)dr+$$

$$\mathbb{E}[c|r_0]\int_0^{r_0}\frac{d}{dr_n}\frac{1}{\sqrt{2\pi}\alpha}\frac{1}{N}\sum_{n=1}^N \exp\left(-\frac{(r-r_n)^2}{2\alpha^2}\right)dr\}\frac{d\tau_0}{1-\tau_0} =$$

$$= \int_0^1 \{\int_{r_0}^1 \frac{1}{\sqrt{2\pi}\alpha^3 N}\mathbf{1}(c_n)\cdot(r-r_n)\cdot\exp\left(-\frac{(r-r_n)^2}{2\alpha^2}\right)dr+ \tag{17}$$

$$\mathbb{E}[c|r_0]\int_0^{r_0}\frac{1}{\sqrt{2\pi}\alpha^3 N}(r-r_n)\cdot\exp\left(-\frac{(r-r_n)^2}{2\alpha^2}\right)dr\}\frac{d\tau_0}{1-\tau_0} =$$

$$= \int_0^1 \{-\frac{1}{\sqrt{2\pi}\alpha N}\mathbf{1}(c_n)\cdot\left[\exp\left(-\frac{(1-r_n)^2}{2\alpha^2}\right) - \exp\left(-\frac{(r_0-r_n)^2}{2\alpha^2}\right)\right]-$$

$$\mathbb{E}[c|r_0]\frac{1}{\sqrt{2\pi}\alpha N}\left[\exp\left(-\frac{(r_0-r_n)^2}{2\alpha^2}\right) - \exp\left(-\frac{(-r_n)^2}{2\alpha^2}\right)\right]dr\}\frac{d\tau_0}{1-\tau_0}$$

Also for the gradients in the code implementation we exploited the trapezoidal rule for the computation of the external integral between [0,1].

## B    ADDITIONAL CALIBRATION METRICS

In this section we are adding results on additional calibration metrics: Kolmogorov-Smirnov (KS) (Gupta et al., 2021), Brier score (Brier, 1950) and class-wise ECE (cw-ECE) (Kull et al., 2019). Results are reported in Tables 5 respectively for CIFAR100, Tiny-ImageNet and DermaMNIST. For all the metrics and datasets, AUCOCLoss provides either the best or comparable results with respect to the baselines.

Table 5: Test results on CIFAR100 for Brier score, class-wise ECE (cw-ECE) and Kolmogorov-Smirnov (KS), both for checkpoint on AUCOC (gray line)and accuracy (white line).

| Loss funct. | Pre TS | | | Post TS | | |
|---|---|---|---|---|---|---|
| | Brier ↓ | KS ↓ | cw-ECE ↓ | Brier ↓ | KS ↓ | cw-ECE ↓ |
| CE | 36,26 | 11,55 | 0,293 | 33,91 | 1,1 | 0,211 |
| | 34,94 | 13,56 | 0,315 | 31,58 | 1,28 | 0,206 |
| FL ($\gamma$=3) | 31,35 | 4,05 | 0,19 | 31,16 | 1,46 | 0,184 |
| | 31,36 | 4,00 | 0,189 | 31,19 | 1,42 | 0,183 |
| AdaFL53 | 31,49 | 4,17 | 0,191 | 31,3 | 1,22 | 0,183 |
| | 31,54 | 3,95 | 0,191 | 31,35 | 1,29 | 0,184 |
| CE+MMCE | 36,15 | 9,99 | 0,271 | 34,6 | 1,23 | 0,209 |
| | 35,07 | 13,25 | 0,311 | 31,87 | 1,46 | 0,199 |
| FL+MMCE | 31,24 | 1,62 | 0,182 | 31,24 | 1,82 | 0,181 |
| | 31,25 | **1,29** | 0,181 | 31,24 | 1,52 | 0,180 |
| CE+Soft-AvUC | 32,29 | 6,82 | 0,215 | 31,71 | 0,53 | 0,199 |
| | 32,42 | 7,14 | 0,219 | 31,76 | 0,62 | 0,197 |
| FL+ Soft-AvUC | 33,34 | 10,05 | 0,258 | 31,74 | 1,58 | 0,177 |
| | 33,43 | 10,23 | **0,179** | 31,78 | 1,8 | 0,176 |
| CE+Soft-ECE | 32,44 | 7,06 | 0,22 | 31,85 | 1,3 | 0,197 |
| | 32,44 | 7,08 | 0,219 | 31,83 | 0,83 | 0,197 |
| FL+Soft-ECE | 32,87 | 6,43 | 0,235 | 32,19 | 1,56 | 0,180 |
| | 32,88 | 6,92 | 0,245 | 32,1 | 1,25 | 0,179 |
| **CE+AUCOCLoss** | 30,54 | 6,72 | 0,21 | **29,78** | **0,75** | 0,184 |
| | 30,62 | 6,3 | 0,204 | 30,0 | 0,93 | 0,188 |
| **FL+AUCOCLoss** | **30,33** | 2,32 | 0,188 | 30,24 | 0,95 | **0,175** |
| | 30,35 | 2,28 | 0,187 | 30,25 | 0,93 | 0,177 |
| **AUCOCLoss Prim.** | 31,01 | 5,93 | 0,242 | 30,49 | 1,52 | 0,185 |
| | 31,14 | 5,95 | 0,245 | 30,62 | 1,59 | 0,188 |

Table 6: Test results on Tiny-ImageNet for Brier score, class-wise ECE (cw-ECE) and Kolmogorov-Smirnov (KS), both for checkpoint on AUCOC (gray line)and accuracy (white line).

| Loss funct. | Pre TS | | | Post TS | | |
|---|---|---|---|---|---|---|
| | Brier ↓ | KS ↓ | cw-ECE ↓ | Brier ↓ | KS ↓ | cw-ECE ↓ |
| CE | 72,56 | 21,38 | 0,287 | 66,25 | **0,74** | 0,163 |
| | 72,89 | 22,05 | 0,295 | 66,14 | 1,05 | 0,160 |
| FL ($\gamma$=3) | 67,29 | 10,34 | 0,198 | 65,91 | 1,14 | 0,152 |
| | 67,42 | 10,73 | 0,198 | 65,93 | 0,74 | 0,151 |
| AdaFL53 | 67,36 | 10,98 | 0,207 | 65,71 | 0,85 | 0,159 |
| | 67,19 | 10,47 | 0,197 | 65,72 | 0,97 | 0,157 |
| CE+MMCE | 72,5 | 21,12 | 0,286 | 66,42 | 0,93 | 0,157 |
| | 72,25 | 20,76 | 0,283 | 66,43 | 0,82 | 0,161 |
| FL+MMCE | 65,53 | 2,7 | 0,158 | 65,49 | 1,63 | 0,151 |
| | 65,67 | 3,38 | 0,158 | 65,53 | 1,05 | 0,151 |
| CE+Soft-AvUC | 72,23 | 21,77 | 0,287 | 65,2 | 1,15 | **0,146** |
| | 72,25 | 21,67 | 0,286 | 65,3 | 0,98 | 0,146 |
| FL+ Soft-AvUC | 65,3 | 7,17 | 0,182 | 64,62 | 1,13 | 0,15 |
| | 65,54 | 8,01 | 0,292 | 64,65 | 1,02 | 0,145 |
| CE+Soft-ECE | 70,34 | 17,56 | 0,251 | 66,14 | 0,81 | 0,15 |
| | 70,34 | 17,56 | 0,251 | 66,14 | 0,82 | 0,150 |
| FL+Soft-ECE | 66,35 | **1,77** | **0,156** | 66,4 | 2,37 | 0,161 |
| | 66,5 | 2,81 | 0,159 | 66,43 | 1,91 | 0,152 |
| **CE+AUCOCLoss** | 66,3 | 12,23 | 0,209 | **64,23** | 1,29 | 0,157 |
| | 66,25 | 12,33 | 0,210 | 64,23 | 1,29 | 0,157 |
| **FL+AUCOCLoss** | 64,54 | 3,13 | 0,165 | 64,45 | 1,29 | 0,155 |
| | **63,65** | 3,11 | 0,164 | 64,47 | 1,24 | 0,158 |
| **AUCOCLoss Prim.** | 66,81 | 8,18 | 0,183 | 65,88 | 2,5 | 0,168 |
| | 66,91 | 8,15 | 0,184 | 65,98 | 2,51 | 0,174 |

# C  CHOICE OF THRESHOLDS $r_0$

The vector of thresholds $\mathbf{r_0} = [r_{0,1}, ..., r_{0,K}]$ is chosen to be equal to the predicted confidences vector $\mathbf{r} = [r_1, ...r_n]$. This choice has a few noticeable consequences. First, it saves time and energy consumption, as the thresholds are inherently learnt from the predicted confidences for each model. Moreover, as $\mathbf{r_0}$ is automatically "calibrated" to the output of each model, the interval of $\tau_0 \in [0, 1]$ is spanned uniformly, providing full flexibility to choose the most suitable operating point. Finally, and most importantly for a feasible implementation of the proposed method as reported in Appendix A.2, this choice makes $\tau_0$ not dependent on the network parameters. In fact, what is relevant for the computation of $\tau_0$ is the proportion of samples whose confidence is smaller than the corresponding $r_0$ and this adaptive choice of thresholds guarantees consistency irrespective of the specific values in $\mathbf{r}$. This can be explained with the aid of Figure 3. The first two axes at the top are the output confidence space for two neural networks (the same conclusion would hold if we considered a single model at two different epochs). The red arrows indicate the smallest output confidence of the dataset for each model, which have two different values. Equally, the green arrows point to the second smallest samples. Given our threshold selection strategy, these samples correspond to the two smallest thresholds for each model, $r_{0,1}^{NN1}$, $r_{0,2}^{NN1}$ and $r_{0,1}^{NN2}$ $r_{0,2}^{NN2}$ for NN1 and NN2 respectively. Irrespective of the different threshold levels between the models, the corresponding values on $\tau_0$ are the same for both NN1 and NN2, respectively 0% and 25% for the first two thresholds $r_{0,1}, r_{0,2}$, in this example with four samples.

Table 7: Test results on DermaMNIST for Brier score, class-wise ECE (cw-ECE) and Kolmogorov-Smirnov (KS), both for checkpoint on AUCOC (gray line)and accuracy (white line).

| Loss funct. | Pre TS | | | Post TS | | |
| --- | --- | --- | --- | --- | --- | --- |
| | Brier ↓ | KS ↓ | cw-ECE ↓ | Brier ↓ | KS ↓ | cw-ECE ↓ |
| CE | 39,71 | 8,06 | 3,22 | 38,11 | 2,15 | 2,22 |
| | 45,48 | 19,83 | 6,01 | 37,01 | 1,45 | 2,53 |
| FL ($\gamma$=3) | 40,00 | 11,14 | 4,06 | 36,55 | 1,77 | 2,03 |
| | 41,68 | 17,02 | 5,12 | 35,69 | 1,81 | 1,91 |
| AdaFL53 | 39,94 | 12,32 | 3,93 | 36,19 | 1,95 | **1,66** |
| | 39,42 | 10,68 | 3,67 | 36,7 | 1,43 | 1,90 |
| CE+MMCE | 41,89 | 10,88 | 3,69 | 38,96 | 2,73 | 2,2 |
| | 44,05 | 18,67 | 5,59 | 36,82 | 2,44 | 2,48 |
| FL+MMCE | 38,96 | **3,10** | 2,55 | 38,91 | 2,29 | 2,34 |
| | 40,71 | 15,74 | 4,72 | 35,73 | 1,06 | 1,89 |
| CE+Soft-AvUC | 38,64 | 5,18 | 2,55 | 38,28 | 2,69 | 2,21 |
| | 45,07 | 18,29 | 5,67 | 38,12 | 3,62 | 2,7 |
| FL+ Soft-AvUC | 39,01 | 6,75 | 3,09 | 37,98 | 2,13 | 1,82 |
| | 39,08 | 12,00 | 3,89 | 36,38 | 2,04 | 1,88 |
| CE+Soft-ECE | 40,37 | 9,39 | 3,31 | 38,39 | 2,73 | 1,92 |
| | 44,67 | 19,53 | 5,87 | 36,7 | 1,55 | 2,15 |
| FL+Soft-ECE | 38,87 | 10,13 | 3,4 | 36,91 | 2,67 | 2,08 |
| | 39,23 | 11,74 | 3,78 | 36,05 | 1,35 | 1,92 |
| **CE+AUCOCLoss** | 38,3 | 7,16 | 4,11 | 36,12 | 1,56 | 1,78 |
| | 38,15 | 12,43 | 3,8 | 35,33 | 1,67 | 1,98 |
| **FL+AUCOCLoss** | **36,07** | 6,55 | 2,55 | **35,46** | 2,04 | 2,04 |
| | 36,02 | 4,06 | **2,27** | 35,71 | 2,11 | 1,93 |
| **AUCOCLoss Prim.** | 37,18 | 5,22 | 2,68 | 38,06 | **1,17** | 2,37 |
| | 36,98 | 4,7 | 2,53 | 36,74 | 3,00 | 2,8 |

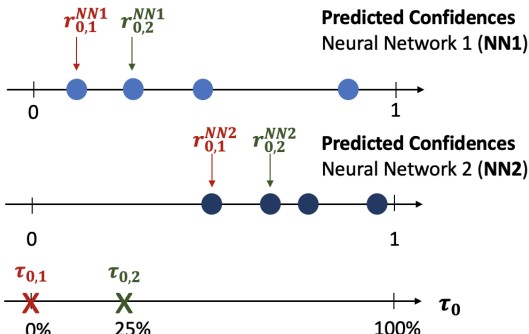

Figure 3: Illustrative example that shows how $\tau_0$ does not depend on any parameter $\theta$ nor on any specific threshold level $r_0$. Even if the models have different threshold levels, the points on $\tau_0$ axis are the same.

# D    RESULTS ON A NEW MEDICAL DATASET: RETINAMNIST

In this Section we report the results on RetinaMNIST (Yang et al., 2021) using ResNet-50 as backbone and batch size 128. The RetinaMNIST is based on the DeepDRiD24 challenge, which provides a dataset of 1,600 retina fundus images. The task is ordinal regression for 5-level grading of diabetic retinopathy severity. Consistently with the original paper, we split the source training set with a ratio of 9 : 1 into training and validation set, and use the source validation set as the test set. For the selection of the common hyperparameters, we followed the suggestions of Yang et al. (2021) for all the baselines and AUCOCLoss.

Table 8: Test results on RetinaMNIST for accuracy, AUCOC and ECE. For each loss function, we report the results of the model check-pointed on the best AUCOC on validation set, pre and post TS. The last two columns report $\tau_0$ corresponding to 65% and 75% accuracy (pre TS). In bold the best result for each metric. The average results over 3 runs are reported.

| | Pre TS | | | Post TS | | $\tau_0 \downarrow$ @ acc. | |
|---|---|---|---|---|---|---|---|
| Loss funct. | AUCOC ↑ | Accuracy ↑ | ECE ↓ | AUCOC ↑ | ECE ↓ | 65% | 75% |
| CE | 70,86 | 51,08 | 13,27 | 72,05 | 11,42 | 49,4 | 63,25 |
| FL ($\gamma$=3) | 69,22 | 50,83 | 17,08 | 69,27 | 11,59 | 51,5 | 76,63 |
| AdaFL53 | 68,84 | 50,75 | 17,38 | 69,21 | 11,58 | 52,88 | 67,00 |
| CE+MMCE | 71,46 | 51,00 | 10,96 | 70,64 | 11,75 | 55,79 | 70,92 |
| FL+MMCE | 70,52 | 51,25 | 10,94 | 70,83 | 10,23 | 56,92 | 83,38 |
| CE+Soft-AvUC | 70,48 | 51,08 | 7,71 | 63,33 | 10,21 | 70,08 | 61,54 |
| FL+ Soft-AvUC | 68,71 | 49,75 | 11,59 | 58,43 | 12,67 | 60,88 | 67,67 |
| CE+Soft-ECE | 72,15 | 52,00 | 9,27 | 70,91 | 11,05 | 49,04 | 62,5 |
| FL+Soft-ECE | 67,99 | 51,83 | 13,44 | 67,9 | 11,05 | 48,33 | 58,96 |
| **CE+AUCOCLoss** | **73,54** | **53,33** | **7,49** | **72,11** | 10,83 | **48,00** | **53,79** |
| **FL+AUCOCLoss** | 71,71 | **53,33** | 11,19 | 70,2 | **9,02** | 48,3 | 69,83 |
| **AUCOCLoss Prim.** | 67,94 | 48,08 | 22,63 | 68,09 | 13,24 | 52,5 | 66,54 |

Table 9: Test results on DermaMNIST for accuracy, AUCOC and ECE for batch size 64 and 32 for models check-pointed on the best AUCOC on validation set.

| | Batch size = 64 | | | Batch size = 32 | | |
|---|---|---|---|---|---|---|
| Loss funct. | AUCOC ↑ | Accuracy ↑ | ECE ↓ | AUCOC ↑ | Accuracy ↑ | ECE ↓ |
| **CE+AUCOCLoss** | 91,16 | 75,18 | 11,00 | 90,14 | 73,8 | 15,5 |
| **FL+AUCOCLoss** | 91,13 | 74,71 | 9,35 | 91,1 | 73,47 | 6,69 |
| **AUCOCLoss Prim.** | 88,85 | 70,31 | 8,9 | 88,68 | 71,19 | 8,36 |

# E    ADDITIONAL RESULTS

In this section we report additional results and plots from our analysis. Moreover Table 9 reports an ablation study on the impact of batch size of both the baselines as well as AUCOCLoss.

Table 10: Test results on CIFAR100 for accuracy, AUCOC and ECE. For each loss function, we report the results of the model check-pointed on the best accuracy on validation set, pre and post TS. The average results over 3 runs are reported. The best results are highlighted in bold.

| Loss funct. | Pre TS | | | Post TS | | $\tau_0 \downarrow$ @ acc. | |
| | AUCOC ↑ | Accuracy ↑ | ECE ↓ | AUCOC ↑ | ECE ↓ | 90% | 95% |
|---|---|---|---|---|---|---|---|
| CE | 82,43 | 78,02 | 13,66 | 93,89 | 4,11 | 23,59 | 38,13 |
| FL ($\gamma$=3) | 93,88 | 77,83 | 3,93 | 93,79 | 1,76 | 24,98 | 40,7 |
| AdaFL53 | 93,87 | 77,77 | 3,95 | 93,78 | 1,74 | 25,14 | 40,54 |
| CE+MMCE | 85,34 | 77,81 | 13,33 | 93,77 | 4,21 | 24,00 | 38,61 |
| FL+MMCE | 93,87 | 77,74 | **1,55** | 93,89 | 1,79 | 25,58 | 40,98 |
| CE+Soft-AvUC | 93,94 | 77,53 | 7,13 | 93,82 | 3,17 | 24,88 | 44,67 |
| FL+ Soft-AvUC | 93,1 | 77,76 | 10,23 | 93,77 | 3,7 | 24,68 | 39,8 |
| CE+Soft-ECE | 93,87 | 77,57 | 7,09 | 93,73 | 3,42 | 24,81 | 40,12 |
| FL+Soft-ECE | 93,42 | 76,8 | 6,92 | 93,59 | 1,43 | 28,08 | 43,38 |
| **CE+AUCOCLoss** | **94,38** | **78,74** | 6,2 | **94,26** | 1,55 | **22,21** | **36,96** |
| **FL+AUCOCLoss** | 94,16 | 78,22 | 2,47 | 94,22 | **1,24** | 23,58 | 38,7 |
| **AUCOCLoss Prim.** | 94,17 | 78,18 | 5,95 | 94,24 | 2,09 | 23,83 | 38,67 |

Table 11: Test results on DermaMNIST for accuracy, AUCOC and ECE. For each loss function, we report the results of the model check-pointed on the best accuracy on validation set, pre and post TS. The average results over 3 runs are reported.

| Loss funct. | Pre TS | | | Post TS | | $\tau_0 \downarrow$ @ acc. | |
| | AUCOC ↑ | Accuracy ↑ | ECE ↓ | AUCOC ↑ | ECE ↓ | 90% | 95% |
|---|---|---|---|---|---|---|---|
| CE | 65,41 | 73,41 | 26,42 | 90,07 | **2,58** | 41,51 | 62,05 |
| FL ($\gamma$=3) | 89,85 | 73,83 | 17,08 | 91,1 | 3,67 | 40,78 | 55,84 |
| AdaFL53 | 89,44 | 73,52 | 10,71 | 90,65 | 2,77 | 42,39 | 58,07 |
| CE+MMCE | 74,89 | 73,25 | 19,13 | 90,43 | 3,84 | 40,4 | 56,74 |
| FL+MMCE | 90,53 | 74,21 | 15,59 | 90,92 | 3,52 | 39,88 | 56,39 |
| CE+Soft-AvUC | 73,27 | 73,64 | 19,04 | 89,98 | 3,9 | 42,54 | 60,47 |
| FL+ Soft-AvUC | 86,78 | 73,18 | 12,02 | 90,84 | 2,77 | 39,42 | 55,59 |
| CE+Soft-ECE | 73,74 | 73,63 | 20,4 | 90,31 | 3,05 | 41,6 | 58,12 |
| FL+Soft-ECE | 89,68 | 73,35 | 11,78 | 90,8 | 3,88 | 41,95 | 56,76 |
| **CE+AUCOCLoss** | 90,59 | 74,78 | 12,47 | 90,9 | 5,8 | 36,63 | 53,15 |
| **FL+AUCOCLoss** | **91,18** | **74,85** | **7,24** | **91,24** | 5,8 | **36,56** | **51,17** |
| **AUCOCLoss Prim.** | 90,55 | 73,8 | 8.3 | 90,57 | 7,9 | 41,7 | 60,77 |

Table 12: Test AUROC(%) on OOD detection for models trained on CIFAR100 and tested on CIFAR100-C (Gaussian noise) and SVHN, pre and post TS. Results correspond to model check-pointed for accuracy.

| Loss funct. | **CIFAR100-C**, AUROC ↑ | | **SVHN**, AUROC ↑ | |
| | Pre TS | Post TS | Pre TS | Post TS |
|---|---|---|---|---|
| CE | 75,75 | 75,91 | 78,9 | 80,41 |
| FL ($\gamma$=3) | 74,94 | 75,14 | 77,37 | 77,88 |
| AdaFL53 | 74,44 | 74,57 | 79,7 | 80,68 |
| CE+MMCE | 75,91 | 75,88 | 75,51 | 76,94 |
| FL+MMCE | 74,17 | 74,11 | 78,28 | 78,11 |
| CE+Soft-AvUC | 73,78 | 73,7 | 77,46 | 78,34 |
| FL+ Soft-AvUC | 75,92 | 75,91 | 79,22 | 80,52 |
| CE+Soft-ECE | 73,32 | 73,19 | 77,17 | 77,93 |
| FL+Soft-ECE | 75,92 | 74,49 | **83,41** | 81,63 |
| **CE+AUCOCLoss** | 75,69 | 76,38 | 81,37 | **82,97** |
| **FL+AUCOCLoss** | 76,08 | 75,83 | 79,03 | 78,24 |
| **AUCOCLoss Prim.** | **78,8** | **78,5** | 79,4 | 78,13 |

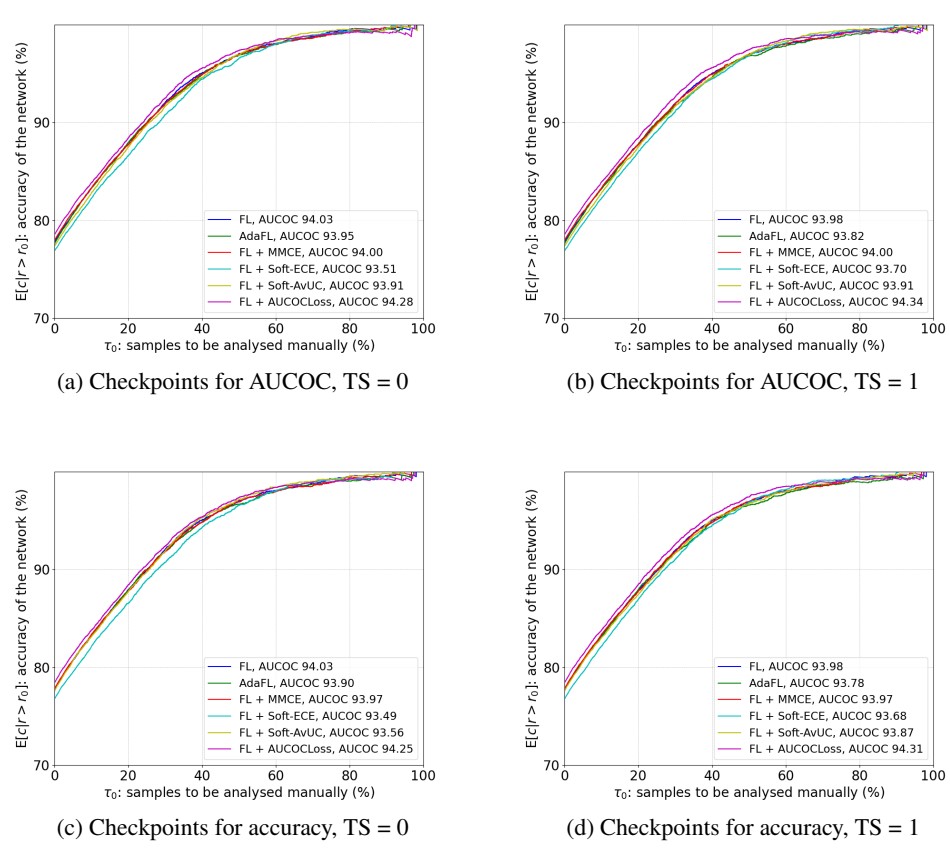

Figure 4: Additional COC curves on CIFAR100. In the first row the plots for models check-pointed with AUCOC, the second with accuracy. In the first column models without TS, in the second with TS.

