# OpenReview forum: "COC curve: operating neural networks at high accuracy and low manual effort"
_ICLR.cc/2023/Conference — Submitted to ICLR 2023_

### Official Review · Reviewer_FvY2 · 2022-10-24

**Confidence:** 3
**Correctness:** 4
**Technical Novelty And Significance:** 3
**Empirical Novelty And Significance:** Not applicable
**Recommendation:** 6

**Clarity, Quality, Novelty And Reproducibility:**

The code is provided and the COC curve, and the AUCOC loss that tries to minimize the expert load are novel.

**Strength And Weaknesses:**

The paper is, in general, well-written.

The authors consider an important problem where the expert load versus the confidence/accuracy of a model is examined.

Taking expert load into account might be helpful in some applications.

Their method seems to have good results both for in and out of distribution cases.

While the toy example is intuitive, the problem is raised because of the binning issue in the ECE metric. Binning-free metrics like the Brier score may not suffer from this issue. Evaluating on other metrics such as Brier, NLL, KS[1], and classwise calibration (calibration on other classes rather than the top one) can further strengthen the paper.

It would be better to discuss the computational time and the extra overhead the proposed method may have in the main paper.
Some details are missing: e.g., the number of bins (M) used for calculating the ECE metric, and the kernel density is computed using batches or the whole dataset.

The ECE errors are sometimes higher than other methods (especially on the DermaMNIST dataset). This brings a few questions: Is AUCOC loss a proper scoring rule? Is this going to have some effect on the results of AUCOC?

Two checkpoints are considered: one by the accuracy of the model and the other by the AUCOC value. Which one do the authors suggest using?

[1] Gupta et al., "Calibration of neural networks using splines," ICLR 2022.

**Summary Of The Paper:**

This paper considers the fact that there should be an operating point based on the confidence of a model (deep neural network) in which an expert should examine the decisions in human-AI collaboration settings. The authors propose a confidence operating characteristics (COC) curve to summarize a model's performance. In COC, the x-axis is the proportion of samples with confidence scores less than a threshold; that is the proportion of samples that should be delivered to an expert. The y-axis is the model's expected accuracy for samples with higher confidence scores than the threshold. The curve is constructed by varying the threshold value. Then, the authors propose the area under the COC curve (AUCOC) that summarizes the curve as a single metric. They use the metric as an objective (usually combined with other metrics such as cross-entropy) to train deep nets. Kernel density estimation is utilized to compute the (extra) terms in the objective. They claim that the trained networks with this extra objective are more accurate and require fewer samples to be delegated to the experts.

**Summary Of The Review:**

I think the paper is well-written and well motivated. The results look good and the code is also provided. A bit more details on the experiments and analysis of the introduced loss function might be required.

---

> ### Author Response · Authors · 2022-11-17
> **Response to Reviewer FvY2**
>
> We thank the reviewer for the comments and questions.
>
> ### **1. Using different calibration metrics.**
>
> Following the reviewer suggestion, we added to the revised Appendix Tables 5, 6 and 7 with KS, Brier score and Class-wise ECE metrics pre and post temperature scaling in order to make the analysis more complete.
>
> ### **2. Additional details.**
>
> We thank the reviewer for pointing out to some details missing. We add them also the paper.
>
> M=15 consistently with the baselines [1, 2].
> KDE is applied batch wise. We also provide results with different batch sizes in table 5 of Supplementary material on the smallest dataset, DermaMNIST, to check whether different batch sizes would impact the proposed loss.
>
> [1] Archit Karandikar, Nicholas Cain, Dustin Tran, Balaji Lakshminarayanan, Jonathon Shlens, Michael Curtis Mozer, and Becca Roelofs. Soft calibration objectives for neural networks. In NeurIPS, 2021.
>
> [2] Jishnu Mukhoti, Viveka Kulharia, Amartya Sanyal, Stuart Golodetz, Philip Torr, and Puneet Doka-nia. Calibrating deep neural networks using focal loss. In H. Larochelle, M. Ranzato, R. Hadsell, M.F. Balcan, and H. Lin (eds.), Advances in Neural Information Processing Systems, volume 33, pp. 15288–15299. Curran Associates, Inc., 2020.
>
> ### **3. Relation with calibration.**
>
> We would like to emphasise that COC curve does not include a measure of calibration and the aim of optimizing for AUCOC is not calibrating neural networks. We compare against calibration methods because 1) there is not a direct literature that we can compare with which optimizes for expert's manual load and accuracy, 2) similar to our method, calibration methods  also aim at improving the interaction between human and AI, by enabling the networks to delegate the decision to human when they are not very confident.
>
> The goal of this paper is maximising AUCOC, i.e. optimizing the trade-off between the accuracy of the network on the analysed samples for different confidence thresholds, against the amount of analysis required from the human expert. This gives domain experts the flexibility to find the most suitable working point on COC.
>
> ### **4. Which checkpoint to use.**
>
> Checkpointing with either AUCOC or accuracy provides comparable results as shown by the results. In general, optimising for AUCOC also should also enforce the optimization of classic accuracy, as on COC it corresponds to the point at x=0%, i.e. zero workload for the domain expert.

---

> > ### Comment · Reviewer_FvY2 · 2022-11-18
> > **Response to Authors -- relation to calibration**
> >
> > I thank the authors for providing more results on different calibration metrics and adding more details about their method.
> >
> > Regarding the relation to calibration, I am aware that the aim is not calibrating neural networks. However, my point is that since the method deals with the amount of expert load based on the **confidence threshold**, a poorly calibrated model trained with AUCOC may result in delegating samples to experts based on wrong **confidence** scores. For example, if the network confidences are 5% less or more than their true probabilities, will this assign less or extra load on the experts than what it should really do? How does this affect the results in the paper? A bit of discussion about the AUCOC being a proper loss or not would also be interesting.
> >
> >
> > I disagree with reviewer 4BfS that the counter-example provided in Fig.1 for ECE also applies to the KS metric. Here are the details:
> >
> > Suppose the confidences of the samples are as follows (from left to right): $0.45, 0.55, 0.65, 0.70, 0.75$. Then, the binary correct (1) and incorrect (0) decisions for networks NN1 and NN2 would be $0, 1, 0, 1, 1$ and $1, 0, 1, 1, 0$, respectively. The KS metric calculates the maximum difference between the cumulative scores and the binary decisions. In this case, the cumulative scores would be $0.45, 1.00, 1.65, 2.35, 3.15$, and the cumulative decisions for NN1 and NN2 are $0, 1, 1, 2, 3$ and $1, 1, 2, 3, 3$, respectively. So, both networks have a $0.65$ KS score.

---

> > > ### Author Response · Authors · 2022-11-18
> > > **Response to Reviewer FvY2**
> > >
> > > We thank the reviewer for the questions and for checking the example proposed!
> > >
> > > We would like to highlight that the exact confidence values are not relevant for the computation of COC, as it is sensitive only to the ranking of the correctly/incorrectly classified samples. Therefore, a practitioner would select as operating point on COC the most suitable trade-off between the performance of the NN on the examined samples and the remaining workload, without taking into consideration the exact value of the confidences.
> > > Calibration is sensitive to the values of predictive confidences (as also shown by the example), while AUCOC to their ordering. As a consequence, AUCOCLoss is not a proper scoring rule for calibration, while it is for AUCOC (solely a model with perfect accuracy would have AUCOC=100% and therefore AUCOCLoss=0).
> > >
> > > Nevertheless, results prove that AUCOCLoss can also provide comparable results to calibration methods, while achieving in general the best AUCOC and accuracy.

---

> > > > ### Comment · Reviewer_4BfS · 2022-12-06
> > > > **Thanks for pointing out**
> > > >
> > > > I thank reviewer YvY2 for pointing this out. I double-checked it and I was mistaken. After reading the authors' responses I'm convinced that AUCOC adds value beyond KS and other calibration metrics.

---

### Official Review · Reviewer_4BfS · 2022-10-25

**Confidence:** 4
**Correctness:** 3
**Technical Novelty And Significance:** 2
**Empirical Novelty And Significance:** 3
**Recommendation:** 6

**Clarity, Quality, Novelty And Reproducibility:**

## Clarity
The paper has some concerns about clarity. For example, what are p(r) and c in eq. 1? Also sec 2.3 is not clear to me. Overall, the clarity of the method section could be improved by explaining the equations in simple words so that they could be understood by non-experts.

## Quality and Novelty
As mentioned previously, there are similarities to the KS metric and it is not clear how much value the presented COC/AUCOC adds to the literature.

## Reproducibility
Please release the code to ensure reproducibility.


**Strength And Weaknesses:**

## Strengths
1. AUCOC is interesting as a measure of a trade-off between confidence and accuracy and also as an auxiliary loss.
2. The experiments show marginal improvements in this trade-off as measured by AUCOC.

## Weaknesses
1. An important paper is not cited [a]. For instance, the COC curve introduced in this paper has similarities to the ones used in [a] and the Kolmogorov-Smirnov (KS) metric has similar properties to AUCOC. Specifically, the counter-example provided in Fig.1 for ECE is not applicable to the KS metric. This raises questions about the value addition of the COC/AUCOC. Please address this.
2. The value of AUCOC as a loss is not very clear to me. Is it improving calibration if so it should be shown in the evaluation. I think it would be better to show the KS metric in table 1 to show calibration. As we know, ECE is susceptible to the binning scheme.
3. Section 2.3 makes multiple approximations and it is not clear about the effect of them in practice.


[a] Gupta K, Rahimi A, Ajanthan T, Mensink T, Sminchisescu C, Hartley R. Calibration of neural networks using splines. ICLR 2021.

**Summary Of The Paper:**

The paper presents a COC curve and area under the COC as a loss to obtain a better trade-off between network confidence and accuracy. The proposed loss together with the standard cross-entropy loss yields a better trade-off as measured by AUCOC.

**Summary Of The Review:**

There are concerns about the value addition of the proposed method and an important reference missing. Due to this I'm recommending rejection but looking forward to the response from the authors.

## Post rebuttal
I thank the authors for the detailed response. I'm convinced that the proposed AUCOC loss adds value beyond calibration and I appreciate the effort to include KS and other calibration metrics. The discussion about the relation to calibration should be included in the paper to improve clarity. I'm increasing the score to marginal-accept.

---

> ### Author Response · Authors · 2022-11-17
> **Response to Reviewer 4BfS**
>
> We thank the reviewer for the comments.
>
> ### **1. Response to comment on novelty.** ###
>
> We thank the reviewer for highlighting an interesting paper like [1], which presents a valuable calibration metrics that we are adding to our evaluation (together with Brier score and Class-wise ECE) in Tables 5, 6 and 7 of the revised Appendix.
>
> However, we would like to explain why [1] is different than the proposed curve and metrics.
> The ultimate goal of COC is to find a trade-off between the accuracy of a NN and the amount of data to be analysed by a domain expert at different operating point.  Instead, the goal of [1] is to minimise expected calibration error. For this purpose, the authors consider the maximal error between the cumulative accuracy and the cumulative score (KS test).  Plots in Figure 1 of [1] are not meant to be optimised, while our goal is to maximise AUCOC.
>
> The closest among the plots in [1] to COC is Fig 1.c, when the authors show percentile VS score: the x axes are the same but the y axes are not. The COC y-axis reports the accuracy over the samples that are classified by the network, i.e. it reports the accuracy on those samples whose confidence is larger than the threshold. Therefore, the normalisation factor for the accuracy changes at every operating point, based on the amount of samples that are left to the network. In COC, y-axis decreases more when deferring a correctly classified sample to the human (penalty - we don't want to defer correctly classified samples from the NN) and it increases when discarding an incorrectly classified sample, which can be assessed by the doctor. On the other hand, in Plot 1.c of [1] if a wrongly classified sample is discarded, the y-axis is not affected at all, as the goal in [1] is focused on improving calibration, therefore not enforcing the same goal COC of optimizing network accuracy VS human workload. Therefore curves in [1] and COC have different effects.
>
> [1] Gupta K, Rahimi A, Ajanthan T, Mensink T, Sminchisescu C, Hartley R. Calibration of neural networks using splines. ICLR 2021.
>
> ### **2. Response to comment about value of AUCOC and use of other metrics.**
>
> We would like to emphasize that COC curve does not include a measure of calibration and the aim of optimizing for AUCOC is not calibrating neural networks. We compare against calibration methods because 1) there is not a direct literature that we can compare with which optimizes for expert's manual load and accuracy, 2) similar to our method, calibration methods also aim at improving the interaction between human and AI, by enabling the networks to delegate the decision to human when they are not very confident.
>
> In our evaluations, we compare our method with the calibration methods using different metrics such as accuracy, AUCOC, amount of samples delegated to expert at a certain accuracy, calibration, and OOD detection. The results demonstrate that AUCOC achieves on par calibration performance with the network calibration methods and improve the other metrics in majority of the cases.
>
> Following the Reviewer suggestion, in the revised version of the paper we added to the revised Appendix Tables 5, 6, 7 with KS, Brier score and Multi-Class ECE as new evaluation metrics.
>
> ### **3. Response to comment about clarity.**
>
> In Section 2.3 we introduce AUCOCLoss, whose objective is to maximise the area of COC (i.e. AUCOC). As shown in Eq. 1 and 2, in order to define the x-y axes of COC it is required to calculate p(r), which is the probability distribution of the predictive confidences (max probability for each sample). However, we do not have any knowledge about p(r); we only have access some samples of it (the output predictive confidences of training samples). In order to estimate the distribution p(r) we applied Kernel Density Estimation (KDE), which is a well-established method to estimate probability distributions from samples, while being directly differentiable. As a consequence of the application of KDE to p(r) (Eq. 5), we also provide the mathematical expression of those terms that depend on p(r) in Eq.6. Note here that r is only one dimensional. Hence, estimation of p(r) does not suffer from dimensionality problem of KDE.
> In the Section 2.1 (called Notation) we defined the notation for the derivations that followed in Section 2.3.
>
> ### **4. Response to comment about reproducibility.**
>
> We would like to point out that the code is already attached as a supplementary material of the original submission.

---

> > ### Author Response · Authors · 2022-11-18
> > **Response to Reviewer 4BfS - 2**
> >
> > ### **5. Additional response to comment 1: KS in the example of Figure 1.**
> >
> > As also noted by reviewer FvY2 with a counter-example, we would like to point out that it is not true that the KS scores of the two models are different. In fact, there exist some confidence configurations where the KS scores of the two models are exactly the same, while the result of AUCOC remains consistent for any configuration of the confidences.

---

### Official Review · Reviewer_DLK2 · 2022-11-01

**Confidence:** 4
**Correctness:** 3
**Technical Novelty And Significance:** 2
**Empirical Novelty And Significance:** 2
**Recommendation:** 3

**Clarity, Quality, Novelty And Reproducibility:**

The proposed method is simple and sufficiently described. However, the method lacks novelty. The experiments seem simple enough to be reproduced.


**Strength And Weaknesses:**

The paper addresses an important but often neglected problem in the field: the search for an operating point in the implementation of an automatic prediction system coupled with a manual validation. This problem is closely related to the search for a good calibration for machine learning models and in particular neural networks.

However, the proposed curve is not new and has been used for a long time in industrial systems.  See for example the read/substitution curves, used to determine the number of rejected bank checks:

 N. Gorski, V. Anisimov, E. Augustin, O. Baret, and S. Maximov, “Industrial bank check processing: the A2iA CheckReader,” Int. J. Doc. Anal. Recognit., pp. 196–206, 2001.

Experiments seem to show an advantage to the proposed method, but this advantage remains limited. The datasets chosen are perhaps not the most adequate to highlight the contribution of the proposed method because, apart from DemaMNIST, they do not correspond to use cases where a minimum performance is fixed and implies manual processing to achieve it. An illustration on real cases of the contribution of the proposed method in terms of reduction of the percentage of manual processing would be interesting. Figure 4 in the appendix gives an indication in this sense and shows that the difference between the different methods is very small.



**Summary Of The Paper:**

This paper focuses on the trade-off between the error rate of the classifier and the number of examples that need to be processed manually because the system was not confident enough to process them automatically. This trade-off is at the heart of all uses of automatic classifiers for industrial tasks with maximum error rate constraints. The authors propose a curve (Confidence Operating Characteristics (COC)) and a measure on this curve (AUCOC) to compare classifiers by taking into account the rate of examples rejected to a treatment because of a lack of confidence of the classifier. The authors compare the use of a loss function derived from this curve for training neural networks with standard calibration methods. The experiments are conducted on 3 datasets (CIFAR100, TinyImageNet and DermaMNIST)



**Summary Of The Review:**

The method proposed in this article is not totally new and its experimental evaluation does not show a major gain compared to other calibration methods. The experiments remain somewhat limited, on data sets that do not really correspond to industrial use cases

---

> ### Author Response · Authors · 2022-11-17
> **Response to Reviewer DLK2 - 1**
>
> We thank the reviewer for the feedback and pointing out a relevant paper.
>
> ### **1. Response to comment about novelty.**
>
> In our submission, we made two claims: 1) proposing the COC curve, 2) proposing a method for optimizing AUCOC.
> The paper pointed out by the reviewer shows that our first claim is not true since the same curve has been used to evaluate the performance of ML models in the past. We unfortunately missed this article despite our best efforts for the literature survey. This article uses very different terminology to the current jargon and that may be the reason. In light of this paper, we gladly removed this claim in the revised manuscript.
>
> However, we still believe that our second contribution is novel. In [1], and some other papers that cite [1], the same curve is used for evaluating the performance of an ML model. The performance of the ML models is considered better when the area under this curve is larger.  Therefore, optimizing ML models with this objective can be beneficial for all the applications that use this metric in their evaluation.  Nevertheless, we still have not seen any work that optimizes an ML model with a similar objective. In our paper, we propose a formulation for the loss function to optimize the area under the COC curve to train neural networks and we show that it significantly helps operating neural networks at high accuracy and low manual effort. The loss function formulation and the idea of optimizing directly this metric is therefore novel to the best of our knowledge.
>
> [1] N. Gorski, V. Anisimov, E. Augustin, O. Baret, and S. Maximov, “Industrial bank check processing: the A2iA CheckReader,” Int. J. Doc. Anal. Recognit., pp. 196–206, 2001.
>
> ### **2. Response to comment about the choice of datasets.**
>
> Having identified the most relevant literature on model calibration, we decided to assess COC/AUCOC on the same datasets employed by the baselines designed for calibration [2, 3], in order to provide consistency.  Both the methods for classic calibration and our work find in the medical field a meaningful real-world application, therefore, in addition to what has been previously done by the SoA, we decided to evaluate the models also on a medical dataset, i.e. DermaMNIST.
>
> Following the suggestion of the Reviewer, we added in Table 8 of the revised Appendix the results on one more medical dataset, RetinaMNIST [4]. The RetinaMNIST is based on the DeepDRiD24 challenge, which provides a dataset of 1,600 retina fundus images. The task is ordinal regression for 5-level grading of diabetic retinopathy severity. We split the source training set with a ratio of 9 : 1 into training and validation set, and use the source validation set as the test set. For the selection of the common hyperparameters, we followed the suggestions of [4] for all the baselines and AUCOCLoss, further investigation might improve the results even more.
>
> Also on this additional dataset we can observe a consistent behaviour as on the other datasets.
>
> [2] Archit Karandikar, Nicholas Cain, Dustin Tran, Balaji Lakshminarayanan, Jonathon Shlens, Michael Curtis Mozer, and Becca Roelofs. Soft calibration objectives for neural networks. In NeurIPS, 2021.
>
> [3] Jishnu Mukhoti, Viveka Kulharia, Amartya Sanyal, Stuart Golodetz, Philip Torr, and Puneet Doka-nia. Calibrating deep neural networks using focal loss. In H. Larochelle, M. Ranzato, R. Hadsell, M.F. Balcan, and H. Lin (eds.), Advances in Neural Information Processing Systems, volume 33, pp. 15288–15299. Curran Associates, Inc., 2020.
>
> [4] Jiancheng Yang, Rui Shi, Donglai Wei, Zequan Liu, Lin Zhao, Bilian Ke, Hanspeter Pfister, and Bingbing Ni. "Medmnist v2: A large-scale lightweight benchmark for 2d and 3d biomedical image classification." arXiv preprint arXiv:2110.14795, 2021.
>
> ### **3. Response to comment about extent of improvement in results.**
>
> We agree with the reviewer that the plot resolution of figure 4 in the Supplementary materials is not visually ideal to appreciate the improvements provided by the proposed loss function.
> However, the last two columns in Tables 1, 2 and 3, which show two examples of operating points on COC, make this more clear. In general, we would like to point out that, while some of the baselines may be close to the results of AUCOCLoss for certain metrics, none of them is consistently close to AUCOCLoss across all the datasets. For example, when looking at the last two columns of Tables 1 and 2, CE+MMCE is worse by 4% than AUCOCLoss on Tiny-ImageNet, but by around 9% in CIFAR100.

---

> > ### Author Response · Authors · 2022-11-17
> > **Response to Reviewer DLK2 -2**
> >
> > ### **4. Response to comment about illustration on real cases.**
> >
> > COC curve and its optimization would be helpful in any task where there is a need to balance the performance of a neural network and the amount of samples delegated to domain expert. Specifically, we identified in the medical image analysis field a valuable example. Radiologists have to analyse a large amount of MRI images every day, as part of the diagnosis process. This task is becoming more and more time consuming, given the continuously growing number of MRI scans required nowadays  [5]. Therefore, based on the specific task, it would be beneficial for a doctor to be able to leave to an automatic system the analysis of those images for which the network is more certain about (guaranteeing a certain performance), while manually inspecting those samples that have lower confidence. A curve like COC allows practitioners to choose flexibly an operating point based on the accuracy expected from the network and their workload.
> >
> > MedMNIST and the newly added RetinaMNIST are two examples of potentially useful applications.
> >
> >
> > [5] https://www.cqc.org.uk/sites/default/files/20180718-radiology-reporting-review-report-final-for-web.pdf

---

> ### Author Response · Authors · 2022-12-06
> **Further concerns to be addressed**
>
> Dear reviewer,
>
> The discussion period is approaching to the end and another reviewer that had similar concerns increased their score based on our rebuttal.
>
> We kindly wanted to engage with you again and ask if you have any further concerns regarding our paper. We would be happy to discuss and address your questions in the remaining time.

---

### Author Response · Authors · 2022-11-17
**General Response**

We thank the reviewers for the suggestions and comments, which we addressed individually.

We modified the paper accordingly to the suggestions:
- Revising the contribution
- Adding Tables in Appendix with KS, Brier score and Class-wise ECE
- Adding results on another medical dataset, RetinaMNIST
- Adding further implementation details

---

### Decision · Program_Chairs · 2023-01-20

**Decision:**

Reject

**Justification For Why Not Higher Score:**

As multiple reviewers pointed out, the experimental validations in this paper are not particularly strong. Coupled with the issues with the novelty of COC, the AC believes that rejection is a suitable choice.

**Justification For Why Not Lower Score:**

N/A

**Metareview: Summary, Strengths And Weaknesses:**

This paper presents a trade-off between the model accuracy and the amount of data needed to be validated by domain experts. In particular, the paper presents 1) Confidence Operating Characteristics (COC) curve and 2) a new loss function for optimizing the area under the curve.

Strength:
- A measure (as well as an auxiliary loss) for computing the trade-off between confidence and accuracy
Generally a well-written paper.
- AUCOC achieves similar calibration performance while improving other metrics in most cases.

Weakness:
- The COC was not new “N. Gorski, V. Anisimov, E. Augustin, O. Baret, and S. Maximov, “Industrial bank check processing: the A2iA CheckReader,” Int. J. Doc. Anal. Recognit., pp. 196–206, 2001. (as pointed out by Reviewer DLK2)
- Limited experimental validation on real datasets.

The authors’ responses address some of the concerns. Reviewer FvY2 and Reviewer 4BfS engaged in discussions with the authors and clarified several misunderstandings (e.g., the relationship with the KS metric). The AC, however, has concerned about the novelty issue brought up by Reviewer DLK2. Given that the main contribution is to "apply" an existing work on COC (which is widely used in the literature) to DL, the authors should discuss why this work has not been used in DL and the associated challenges. Then articulate *how* they address the challenge (which will be the main contribution). After reading all the reviews and discussions, the AC is convinced that the paper, in its current state, does not have sufficient merits and recommends to reject.


**Summary Of Ac-Reviewer Meeting:**

N/A